# Subsurface manifestation of Marine Heatwaves in the South West Indian Ocean

Clea B. Welch[1,2], Neil Malan[3], Daneeja Mawren[1,2], Tamaryn Morris[2], Janet Sprintall[4], Juliet C. Hermes[1,2]

[1]Ocean and Atmosphere Science, University of Cape Town, Cape Town, 7700, South Africa
[2]South African Environmental Observation Network Egagasini node, Cape Town, South Africa
[3]Climate Change Research Centre and Centre of Marine Science and Innovation, University of New South Wales, Sydney, New South Wales, Australia
[4]Scripps Institution of Oceanography, University of California, San Diego, La Jolla, CA, United States

*Correspondence to*: Clea B. Welch (wlccle001@myuct.ac.za)

**Abstract.** Marine heatwaves (MHW) are extreme events of prolonged, anomalously warm ocean temperatures. Globally, marine heatwaves are increasing in frequency and intensity and are responsible for long-term impacts on marine ecosystems, which have devastating socio-economic consequences. A key gap in our understanding of MHWs is how they manifest in the subsurface. This paper uses satellite sea surface temperature (SST) data and *in situ* subsurface temperature observations from Expendable Bathythermographs (XBTs) to investigate the anomalous water temperature characteristics associated with surface identified MHWs in the South West Indian Ocean (SWIO) and how they progress through the water column. We find that the eddy corridors through the SWIO, where EKE is high and SST variability is low, are primarily characterised by the occurrence of abrupt and intense MHWs, and that the frequency, duration and intensity of these events are largely associated with mesoscale activity. In these eddy corridors, surface–detected MHW case studies demonstrate a strong, deep–reaching subsurface temperature anomaly signal with maximum intensity below the surface. The majority of these MHWs are associated with anticyclonic eddies, which provide a possible mechanism for the deep extent of the surface MHWs. Improving our understanding of the interaction between mesoscale features and subsurface MHW characteristics will benefit prediction of MHWs and management of the regions' biodiversity..

## 1 Introduction

Marine heatwaves (MHWs) are extreme, anomalously warm ocean events that are known to have devastating impacts on marine species, ecosystems, and ultimately coastal countries' socioeconomics that depend on a blue economy (Mills et al., 2013; Hobday and Pecl, 2014; Hermes et al., 2019). The thermal stress during MHWs has initiated coral bleaching events, destroyed marine foundation species, caused mass mortality events, species redistributions and resulted in irreversible physiological damage to marine life (Mills et al., 2013; Frölicher et al., 2018; Oliver et al., 2021; Perez et al., 2021; Garrabou et al., 2022; Mawren, et al., 2022 b). The adverse effects of MHWs are especially concerning as, under scenarios of continued

global ocean warming, MHWs are projected to increase globally, with events lasting longer and intensifying. If this trend continues many parts of the ocean are predicted to reach a near-permanent MHW state by the late 21st century (Hobday et al., 2016; Frölicher et al., 2018; Oliver et al., 2018; Holbrook et al., 2020). This highlights the need for rapid improvement in our understanding of MHWs and how to manage or adapt to their impacts (Elzahaby and Schaeffer, 2019).

A key gap in our understanding of MHW events is how they manifest in the subsurface (Oliver et al., 2018). MHW detection and characterization is largely limited to the surface due to the lack of continuous, long-term, and high-resolution subsurface temperature records (Elzahaby et al., 2021). However, MHWs themselves are not surface trapped, as they can penetrate to considerable depths, or even exist at depth with no surface signal (Elzahaby and Schaeffer, 2019; Holbrook et al., 2020; Scannell et al., 2020; Elzahaby et al., 2021; Schaeffer, Sen Gupta and Roughan, 2023; Zhang et al., 2023). The lack of subsurface MHW characterisation limits our understanding of the true impacts these events have, as it is the vertical extent of MHWs that directly impact marine ecosystems (Elzahaby and Schaeffer, 2019; Holbrook et al., 2020; Scannell et al., 2020).

The subsurface extent of MHWs is often associated with different spatial MHW patterns and drivers compared to the surface (Elzahaby and Schaeffer, 2019; Scannell et al., 2020; Perez et al., 2021; Fragkopoulou et al., 2023; Zhang et al., 2023). Different ocean dynamical processes, such as large-scale circulation, oceanic planetary waves, boundary currents, eddies, local downwelling, seasonal stratification and mixing influences the vertical structure of MHWs (Schaeffer, Sen Gupta and Roughan, 2023; Zhang et al., 2023). Upper ocean MHWs (0-150 m) have been shown to mostly originate from anomalous air-sea fluxes, but even in cases where air-sea heat fluxes are the predominant driver of MHWs, ocean dynamical processes can extend the warm MHW signature into the subsurface (Schaeffer, Sen Gupta and Roughan, 2023; Zhang et al., 2023). Deeper MHWs (that extend below 150m) are shown to be mostly driven by deep warm-core, anticyclonic mesoscale eddies (Schaeffer and Roughan, 2017; Elzahaby and Schaeffer, 2019; Perez et al., 2021; Fragkopoulou et al., 2023), boundary current shifts (Großelindemann et al., 2022) and Ekman pumping (Hu et al., 2021). In particular, eddies in subtropical western boundary currents (WBCs) are known to drive deeper and longer-lasting MHWs (Schaeffer and Roughan, 2017; Elzahaby and Schaeffer, 2019; Benthuysen et al., 2020; Elzahaby et al., 2021; Zhang et al., 2023).

Here, the focus is on the South West Indian Ocean (SWIO), which is part of the greater western Indian Ocean global warming hotspot (Roxy et al., 2014), and is host to a highly unique, complex and variable WBC current system that supports one of the six primary global marine biodiversity hotspots (Ramirez et al., 2017). This makes assessing MHWs in this region even more crucial and urgent, yet to date, MHW characterization in this region is sparse and limited to the surface (Mawren et al., 2022 a).

58

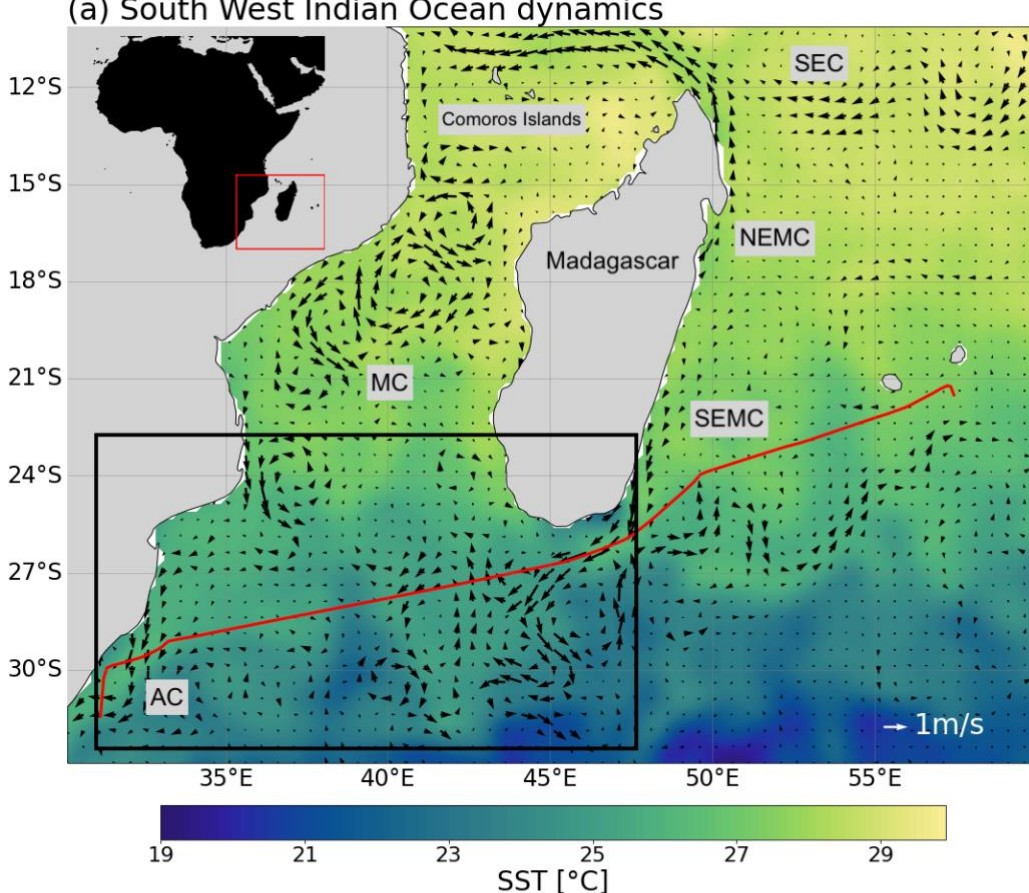

## (a) South West Indian Ocean dynamics

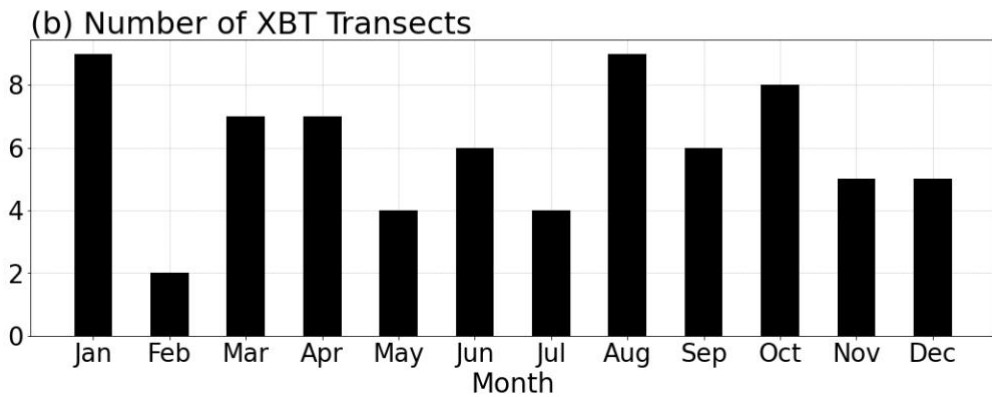

## (b) Number of XBT Transects

59

**Figure 1: (a)** A snapshot of sea surface temperature conditions on 13/04/2021 (OISST V2) of the South West Indian Ocean, with geostrophic current velocities overlaid (satellite AVISO altimeter). Key ocean circulation features are highlighted - SEMC South-East Madagascar Current, NEMC North-East Madagascar Current, SEC South Equatorial Current, Mozambique Channel and Agulhas Current.The black box 22° S - 32° S and 31° E - 48° E) indicates the study area. The red line represents the IX21 HR-XBT transect from Durban, South Africa to Port Louis, Mauritius. **(b)** The number of XBT transects available for each month from the IX21 transect for the period from 1994 – 2022.

A key feature in the SWIO is the Greater Agulhas Current System which consists of the eddying flow through the Mozambique Channel (MC) and two dynamic WBCs, the South East Madagascar Current (SEMC) and the Agulhas Current (AC), the largest and strongest Southern Hemisphere WBC, which plays a vital role in global thermohaline circulation (Fig. 1; Beal et al., 2020). Flow through the MC is characterised by large, deep-reaching, mostly anticyclonic and anomalously warm southward propagating mesoscale eddies, which are formed from baroclinic instability in the South Equatorial Current (SEC), around the Comoros Islands (Collins et al., 2012; Halo et al., 2014; Voldsund et al., 2017). South of Madagascar, the SEMC retroflects and sheds pairs of counter-rotating mesoscale eddies which converge with eddies from the MC (Voldsund et al., 2017). The persistence of eddies from the MC and SEMC creates a state of instability, which is the main source of variability and SST anomalies in the region, making this region especially susceptible to the occurrence of MHWs (DiMarco et al., 2000; Halo et al., 2014; Phillips et al., 2021; Mawren et al., 2022 a).

Where the MC and SEMC converge, eddy activity is highest, and surface-identified MHWs are increasing in frequency, intensity and duration more rapidly than anywhere else in the MC (Mawren et al., 2022 a). Intensification of mesoscale eddies, has subsequently increased ocean warming trends and SST anomalies, which may largely explain these observed MHW trends (Wu et al., 2012; Schaeffer and Roughan, 2017; Benthuysen et al., 2020; Mawren et al., 2022 a). The most intense and longest-lasting MHW in the MC occurred in February 2017 (Mawren et al., 2022 a). The 2017 MHW event was found to be modulated by horizontal advection and the presence of mesoscale eddies, with maximum temperature anomalies peaking when the core of an anticyclonic eddy passed through (Mawren, Hermes and Reason, 2022 a). This suggests that, as seen in other WBC regions (Bian et al., 2023), mesoscale eddies significantly influence the occurrence and intensity of MHWs in the SWIO (Mawren et al., 2022 a). Yet, the exact role mesoscale eddies play in driving surface MHW characteristics and their subsurface extent, remains unclear. This is of particular concern as the SWIO supports a variety of temperature-sensitive, pristine ecosystems with high biological diversity, high endemism and endangered species. The disruption to the ecosystem can have significant socioeconomic impacts, as neighbouring east African countries rely heavily on these ecosystems for fish stocks and marine ecotourism (Obura, 2012; Pereira et al., 2014).

This study aims to build on the limited work that has previously explored MHW events and trends in the SWIO with a focus on their subsurface characteristics. In Section 2, data sets and methodology are described. Section 3 investigates surface MHW characteristics, then, using temperature profiles from in situ Expendable Bathythermographs (XBTs) observations, the subsurface extent of surface-identified MHWs are explored. This provides the first description of subsurface characteristics of MHWs in the SWIO. Particular focus is also placed on understanding the role mesoscale eddies play on both the surface and subsurface extent of MHWs. For this reason, the study area is confined to 22° S - 32° S and 31° E - 48° E, which encompasses the region of greatest ocean variability and mesoscale eddy activity south and southwest of Madagascar (Fig. 1). A Discussion follows in Section 4 with Conclusions in Section 5.

## 2 Methods

### 2.1 Sea surface temperature and surface marine heatwave identification

High resolution gridded (0.25°) NOAA optimally interpolated sea surface temperature (OI SST) V2 data was used to explore SST conditions and identify surface MHWs in the SWIO region from 01/01/1993 – 31/12/2022. This time period matches the same time period of available XBT data, used to investigate the subsurface extent of surface–identified MHWs. OISST data has been widely used to identify and characterise global and regional MHW events and trends globally (Reynolds et al., 2007; Banzon et al., 2016; Frölicher et al., 2018; Sen Gupta et al., 2020; Guo et al., 2022; Saranya et al., 2022) and in the SWIO (Mawren et al., 2022 a; Mawren et al., 2022 b).

MHWs were identified and quantified in the SWIO using the Hobday et al., (2016) definition, which defines a MHW as a discrete, anomalously warm water event, with temperatures that exceed the 90th percentile (the threshold) of the 30-year historical baseline period and have a duration of at least five consecutive days. For consistency with previous studies, a fixed climatological baseline, 1993–2022, and a 31 day smoothing window was used to identify surface MHWs (Smith et al., 2025). The properties of MHWs over the region were described by set metrics: mean duration (the time, in days, between the start and end of a MHW), mean frequency (the number of events that occurred during a year or season), mean intensity (the average temperature anomaly over the duration of the event) and the cumulative intensity (itegrated temperature anomaly for the duration of the event). Seasonal MHW patterns were also investigated using the metric cumulative intensity, which provides a good description of the severity of MHW events (Mawren et al., 2021). Detected MHW events over the entire climatological mean were grouped by austral season (Summer–December, January and February; Autumn–March, April and May; Winter–June, July and August and Spring–September, October and November).

### 2.2 Subsurface temperature measurements from *in situ* data

The subsurface expression of surface MHWs (detected from OISST data) was investigated using XBT data from the near-repeat IX21 HR-XBT transect. XBTs provide temperature profiles from 0 to 850 m depth. IX21 is nominally from Durban, South Africa (29.9° S, 31.0° E) to Port Louis, Mauritius (20.2° S, 57.5° E) (Chandler et al., 2022), but for this study, the portion of the IX21 transect from Durban to south of Madagascar was used.

The IX21 HR-XBT transect is nominally occupied 4 times a year and has a horizontal resolution of 6 – 10 km between XBT profiles within the boundary current and 20–30 km offshore (Goni et al., 2019; Chandler et al., 2022). The transect takes three days to complete and so is considered synoptic. Each transect is assigned the date of the first temperature profile acquired for the cruise. Temperature was objectively mapped to 10m depth intervals from 0m to 800 m and 0.1° intervals in longitude (Chandler et al., 2022). Data was available from 8 September 1994 to 6 September 2022. Since XBT data from the IX21 transect is collected nominally 4 times a year, there is not enough data to produce a high-resolution climatological baseline period, which is required to identify MHWs using the Hobday et al., (2016) definition. Instead, subsurface temperature anomalies calculated relative to the seasonal climatological means, calculated using the XBT

data (Fig. 1b), from 1994 – 2022 were used as the measure of the subsurface extent of surface MHWs, an approach commonly
used when studying subsurface MHW signals using in situ data (Schaeffer and Roughan, 2017; Elzahaby and Schaeffer, 2019;
Elzahaby et al., 2021; Perez et al., 2021). Each seasonal mean consisted of ~15–20 XBT transects (Fig. 1b). To calculate
temperature anomalies, the seasonal climatological means were subtracted from the daily temperature profiles for each data
available day.

**2.3 Sea level anomaly data**

To investigate the influence  of mesoscale eddies on the properties of surface-identified MHWs, high resolution (0.25°),
optimally integrated, gridded daily sea level anomalies (SLA) and geostrophic currents, over a thirty–year period (01/01/1993
– 31/12/2022) were extracted from altimeter satellite data distributed by AVISO (Archiving, Validation, and Interpretation of
Satellite Oceanographic data). Altimeter satellite gridded SLA are computed with respect to a twenty–year [1993, 2012] mean.
This data has previously been used to track eddies during MHWs in the SWIO by Mawren et al.  (2022a). In the SWIO, where
in situ observations are sparse, satellite altimetry provides useful information about mesoscale ocean variability and is the best
way to identify the presence of mesoscale eddies (Halo et al., 2014). Positive (negative) SLA associated with  anticyclonic
(cyclonic) geostrophic currents indicate the presence of warm-core (cold-core) eddies (Halo et al., 2014).
Mean eddy kinetic energy (EKE) for the entire time period was calculated using SLA and geostrophic velocity and used as a
measure of eddy activity across the region. Mean EKE was calculated as:
$$EKE = \frac{1}{2}(u'^2 + v'^2), \tag{1}$$
where $u'^2$  represents the mean squared anomaly of the horizontal velocity component u from its spatial mean and  $v'^2$ represents
the mean squared anomaly of the vertical velocity component v from its spatial mean (Bai et al., 2024).

**2.4 Investigation of subsurface anomaly signals associated with surface MHWs**

For each day when subsurface data from the IX21 XBT  transect were available, and a MHW signal was detected at the
corresponding location, the associated subsurface warm temperature signal was isolated. These subsurface warm temperature
anomaly profiles were used to investigate the subsurface manifestation of surface identified MHWs. The corresponding surface
SLA for each anomaly profile was also recorded to further investigate the influence of mesoscale eddies on the characteristics
of the subsurface anomaly. By correlating the SLA with the temperature anomaly profiles, the analysis aimed to uncover
potential links between mesoscale ocean dynamics and the vertical extent, intensity, and structure of subsurface temperature
anomalies associated with surface MHWs. This dual focus on both subsurface and surface parameters provides a
comprehensive framework for investigating the complex interactions driving the subsurface expression of marine heatwaves.

## 3 Results

### 3.1 Surface MHW variability

The SWIO is defined by distinct spatial patterns that characterize MHW related variability (Fig. 2c–f) that are underpinned by pre–existing mean SST conditions and variability (Fig. 2). On the other hand, despite distinct seasonality in mean SST and variability, MHWs do not exhibit strong seasonality, as the spatial pattern of MHW cumulative intensity remains similar during each season (Not shown). Due to the lack of observed MHW seasonality, the mean MHW trends and their relationship with SST variability and mesoscale eddies are further investigated.

The SWIO experiences warm mean ocean temperatures, up to 28 ℃ , in the Mozambique Channel (21° S) that decrease southward, reaching a minimum of 19 ℃ at 33° S (Fig. 2a). The SST standard deviation shows that, on average, the majority of the region experiences SSTs that exceed the mean by between 1.5–2.2 ℃ (Fig. 2b). On average, surface MHWs occur 1 to 3 times per year, last 8–18 days and reach intensities of 1.0–2.0 °C, but have varying spatial characteristics (Fig. 2b – f). These observed mean values of MHW frequency, duration, intensity and cumulative intensity, are consistent with previous findings by Mawren et al. (2021) in the SWIO, and global WBC studies by Oliver et al. (2018) and Holbrook et al. (2019). Within the southern MC, MHWs occur most frequently (between 2–3 times on average per year), but are shorter-lived (8–12 days) and less intense (1.2–1.5 °C), whereas, in the southeast open ocean region, MHWs occur less frequently (between 1–2 times on average per year), but are longer-lasting (12–18 days)  and more intense (1.4–2.2 °C) (Fig. 2c,d). Similarly, since cumulative intensity is the integration of duration and intensity, MHWs are the most severe (16–19 ℃days) in the southeast open ocean, and least severe in the southern MC (8–16 ℃days) (Fig. 2f).

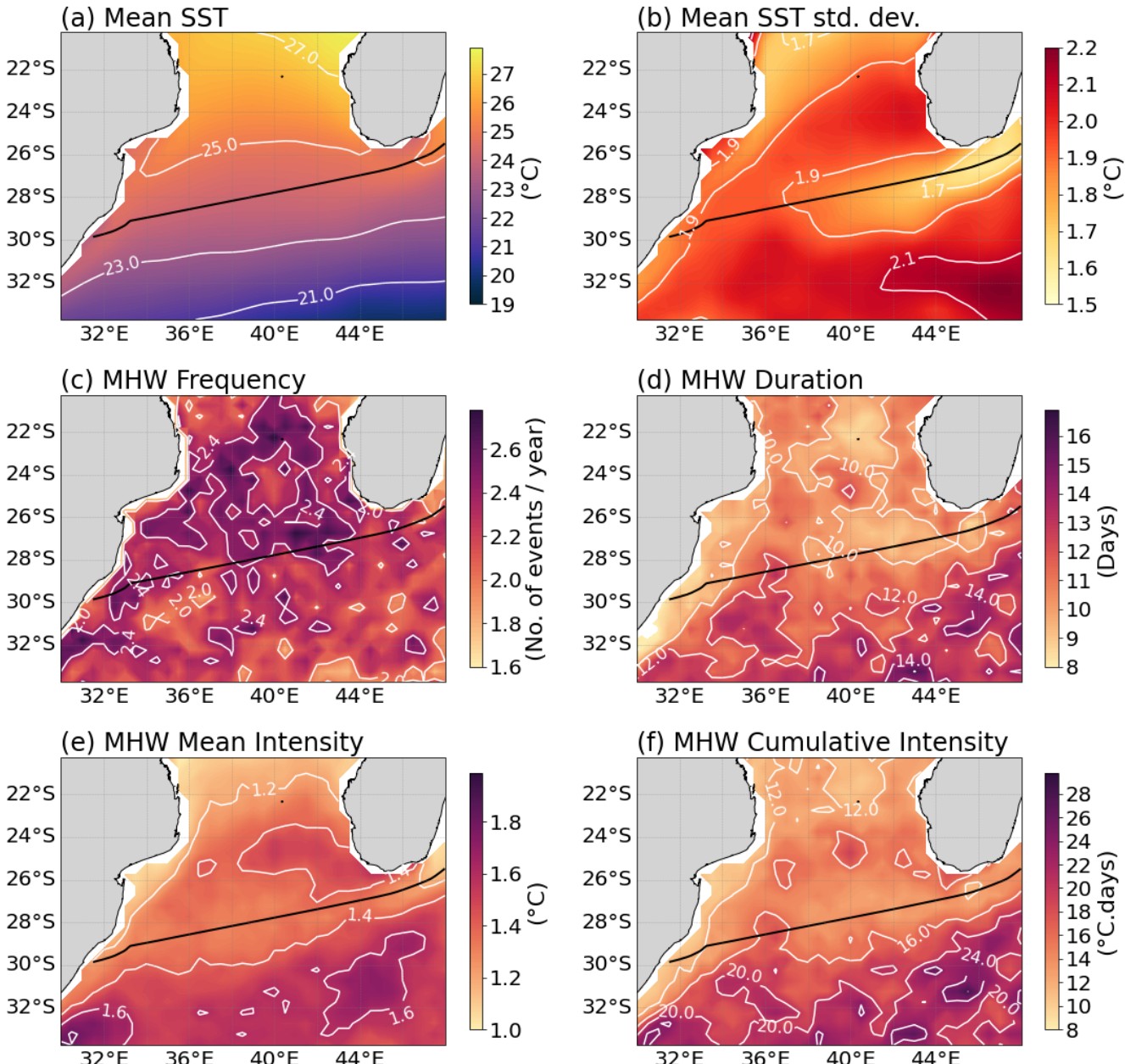

**Figure 2:** Climatological mean **(a)** SST (°C) and **(b)** SST standard deviation (°C). Surface mean annual MHW **(c)** frequency, **(d)** duration (days), **(e)** mean intensity (°C) and **(f)** cumulative intensity (°Cdays) calculated from NOAA optimally interpolated sea surface temperature (OISST V2) high resolution (0.25°) gridded SST data for the climatological period from 1993–2023. The black solid line in all the figures denotes the IX21 XBT transect.

The mean annual spatial distribution of MHW characteristics is closely linked to the underlying patterns of SST variability. Regions with higher SST variability are associated with more intense MHW intensity, as evidenced by a strong correlation (r = 0.75, p–value > 0.05) (Fig. 3b). SST variance is highest off the west coast of Madagascar (1.5 °C) and in the southern–most

open ocean (1.9–2.2 °C) but is weakest along the west coast of Africa and at the region of SEMC leakage (Fig. 3b), confirming the spatial relationship between SST variance and MHW intensity (Fig. 2e, Fig. 3b).Weaker SST variance within the southern MC and along the west coast of Africa, compared to the southernmost open ocean region, also closely resembles the patterns of MHW duration and cumulative intensity, which suggests that longer (shorter)-lasting and more (less) severe MHWs are associated with higher (lower) SST variance (Fig. 2b, d, f). Unlike MHW duration, intensity and cumulative intensity, the mean spatial distribution of MHW frequency reflects an inverse pattern compared to SST variance, suggesting that where SST variance is high (low), MHWs occur less (more) often (Fig. 2b, c).

To investigate the role that underlying oceanographic processes play in driving the observed mean surface MHW characteristics, we investigate the relationship between EKE, geostrophic velocities and SST variability (Fig. 3a). The westward–flowing SEMC leakage is visible, as is the train of eddies flowing southward through the MC and the fast, southward–flowing AC (Fig. 1a and 3a). EKE is strongest on the western side of the MC and between the southern tip of Madagascar and the continent, where stronger poleward and south–westerly currents are present, respectively (Fig. 3b).

Furthermore, the significant correlation between SST variability and MHW intensity is closely linked to EKE (Fig. 3b). Regions with high EKE (0.08–0.16 m²/s²) exhibit low SST variability and are characterized by frequent, short–lived, and less intense MHWs. This suggests that high EKE, driven by the mean surface flow, promotes the formation of MHWs but also accelerates their dissipation due to shorter residence times. Conversely, regions with low EKE (0.0–0.08 m²/s²) experience higher SST variability and support less frequent but more intense and longer–lasting MHWs. This implies that reduced EKE allows for more stable thermal conditions, fostering the persistence of intense MHW events (Fig. 2b, c; Fig. 3). This observed relationship reinforces the role of local thermal fluctuations in driving extreme ocean warming.

202

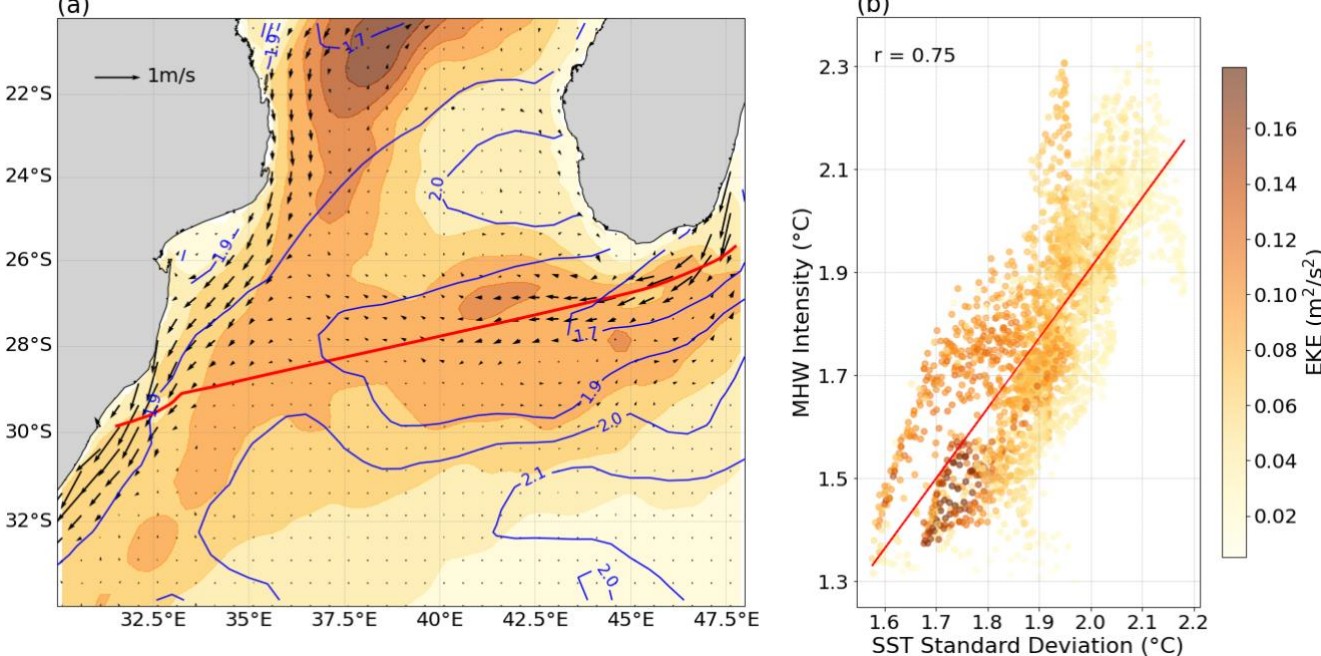

203

**Figure 3: a)** Mean eddy kinetic energy (EKE), with mean geostrophic velocities overlaid, calculated from high resolution AVISO data for the climatological period from 1993–2023. The blue contour lines indicate SST standard deviation (°C) calculated from NOAA optimally interpolated sea surface temperature (OISST V2) high resolution (0.25°) gridded SST data for the same period from 1993–2023. The red solid line denotes the IX21 XBT transect. **b)** Scatter plot of mean annual MHW Intensity (°C) and SST standard deviation (°C). Colours indicate the EKE at these points. The regression line (red) between MHW Intensity and SST standard deviation and the significant correlation coefficient are indicated.

### 3.2 Subsurface temperature anomaly properties associated with surface–identified MHWs

To examine the subsurface signal of MHWs, we identified when surface MHWs (detected from SST satellite data) were both co–located with and occurred simultaneously along the IX21 XBT line. MHWs of varying intensities and sizes were identified in the SST data on 65 days with corresponding XBT transects. For each day and location where a MHW signal was present over the XBT transect, 92% experienced maximum temperature anomalies below the surface (0 m) and 68% of the subsurface warm anomaly profiles (associated with the presence of MHWs), extended down to 800 m.

A significant relationship was found between surface temperature anomalies and maximum subsurface anomalies ($r = 0.70$, p–value $< 0.0001$), as subsurface anomalies tend to increase with increasing surface anomalies (Fig. 4a). Furthermore, deeper MHW events (depth of subsurface anomaly maxima $> 500$ m) tend to experience stronger subsurface anomalies than shallower events in the upper 200 m where surface and subsurface anomalies are more comparable (Fig. 4a). This observed depth–dependent pattern suggests that the depth of the subsurface warm anomaly may play a role in modulating subsurface thermal responses to surface anomalies.

According to the distribution of surface and subsurface anomalies, there is a significant difference between the surface and subsurface anomalies ( p–value < 0.001) (Fig. 5b). Subsurface anomalies are, on average, 1.04 °C warmer than the anomalies at the surface and experience a larger range of warmer temperatures than that of surface anomalies (Fig. 4b). These differences in mean values and variability indicate that temperature anomalies, associated with surface MHWs, intensify below the surface.

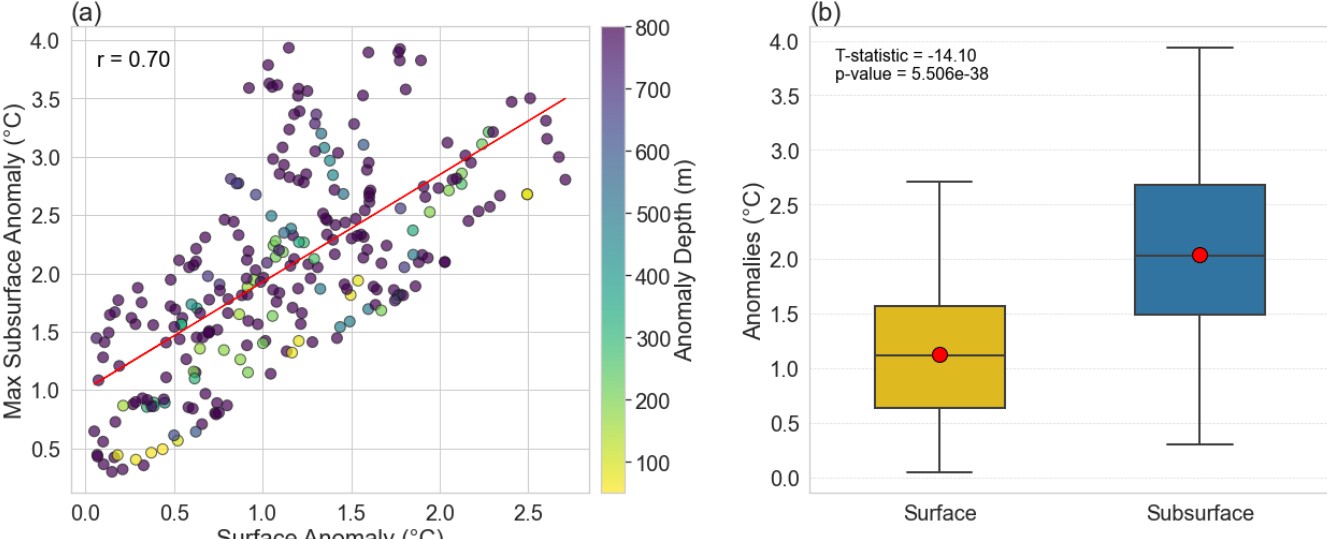

**Figure 4: a)** Scatter plot of surface anomalies and maximum subsurface anomalies at points where surface MHWs were identified. Colours indicate the maximum depth of warm temperature anomalies. The regression line (red) between surface and subsurface anomalies and correlation coefficient are indicated. **b)** Boxplot comparing surface and subsurface anomalies. Red markers indicate mean values, while the central line shows the median. Boxes represent the interquartile range (IQR), with whiskers extending to 1.5 times the IQR. T–statistic and p–value from a two–sample t–test are displayed, highlighting the significance of the difference.

To further explore subsurface intensification of the temperature anomaly signals associated with MHWs, the relationship between maximum subsurface temperature anomalies (°C) and their corresponding depths (m) was evaluated in relation to sea level anomalies (SLA) (Fig. 5). A positive, statistically significant relationship between maximum subsurface anomaly temperature is found, with a correlation of 0.5 (p–value < 0.0001), showing that the warmest subsurface temperatures are typically associated with greater depths (Fig. 6a). This highlights a positive association between the magnitude of the subsurface temperature anomaly and the depth at which it occurs and points to an underlying physical or environmental mechanism linking these variables.

Since the SWIO is an eddy–dominated region, the influence of mesoscale eddies on the depth of maximum subsurface anomalies is investigated. The majority of the subsurface temperature anomaly profiles are associated with anti–cyclonic eddies, with 78.62% of the profiles associated with positive SLA (Fig. 5a). These subsurface maximum temperature anomalies occur at depths ranging from 0–270 m, whereas profiles associated with cyclonic eddies experience maximum temperature anomaly depths ranging from 0–60 m, with two outliers extending to depths of 170 m and 180 m (Fig. 5b). Furthermore the mean maximum subsurface anomaly associated with anti–cyclonic eddies (positive SLA) is significantly deeper than the mean

maximum subsurface anomaly depth associated with cyclonic eddies (p–value < 0.0001), with mean depths of 100 m and 51
m respectively (Fig 5b). This highlights the significant role of mesoscale eddies in shaping the vertical distribution of
subsurface temperature anomalies in the SWIO, with anti–cyclonic eddies driving temperature anomalies to greater depths
compared to the shallower depths associated with cyclonic eddies.

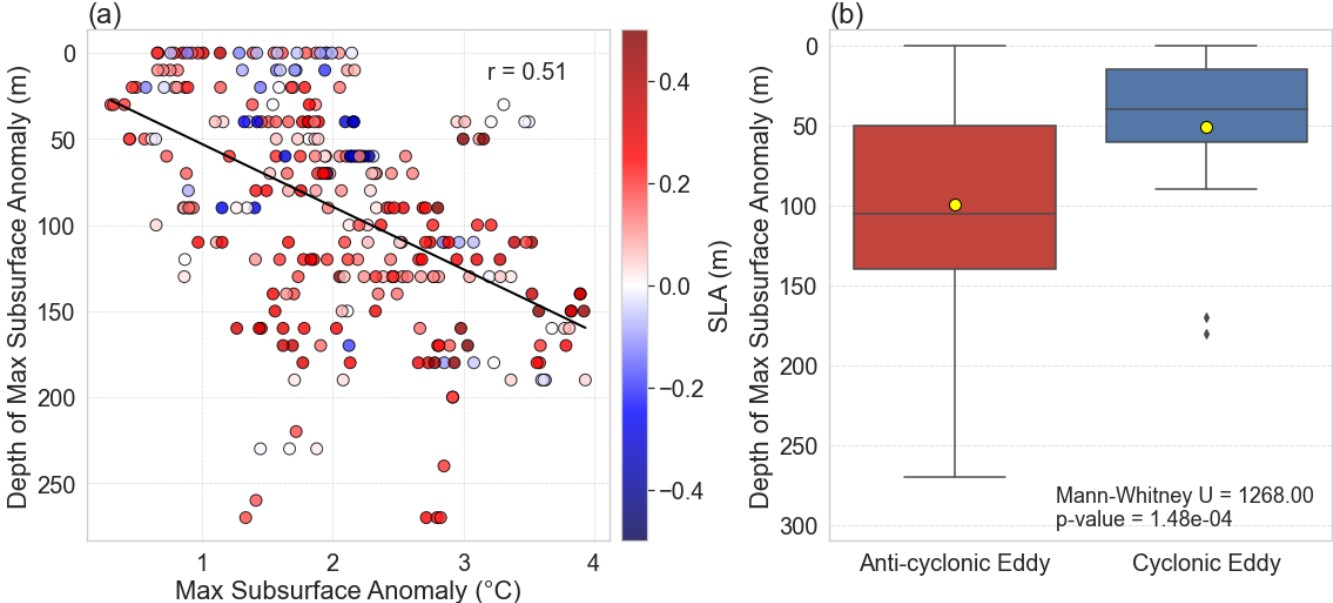


**Figure 5: a)** Scatterplot showing the relationship between maximum subsurface temperature anomalies (°C) and their corresponding depths
(m), color–coded by Sea Level Anomalies (SLA, m). Positive SLA values (red) indicate anti–cyclonic eddies, while negative SLA values
(blue) represent cyclonic eddies. **b)** Boxplot comparing the depths of maximum subsurface anomalies between anti–cyclonic and cyclonic
eddies. Yellow markers indicate mean values, while the central line shows the median. Boxes represent the interquartile range (IQR), with
whiskers extending to 1.5 times the IQR. Mann–Whitney U and p–value from a two–sample t–test are displayed, highlighting the significance
of the difference.

**3.3.1 Case studies of MHW events in the SWIO and their subsurface manifestation**

Given the statistically significant relationship between surface MHW signals and subsurface temperature anomalies, as well
as the depth–dependent structure of these anomalies in relation to mesoscale eddies, three surface–identified MHW events (9
January 2020, 14 July 2012 and 20 October 2007) were selected as case studies. These events, which persisted over the IX–
21 XBT transect, were analysed to further investigate their spatial and subsurface characteristics in the SWIO (Fig. 6).
For all three case studies, the spatial distribution of the MHWs are aligned with the spatial distribution of the warm–core
anticyclonic eddies, with maximum MHW intensities directly surrounding the cores of the largest anticyclonic eddies (Fig.
6a–d). For example, for the 9 January 2020 MHW case study, the maximum MHW intensities (2.5 °C) are identified
surrounding the largest anticyclonic, with warm core eddies of 0.5 m above sea level, at 27–32° S and  36–42° E (Fig. 6a and
d). Likewise, on 14 July 2012 and 20 October 2007, the most intense surface MHW temperatures were recorded (2 °C and 1.5
°C, respectively) surrounding the largest sea level anomalies (0.4–0.5 m above sea level) (Fig. 6b,c,e and f).

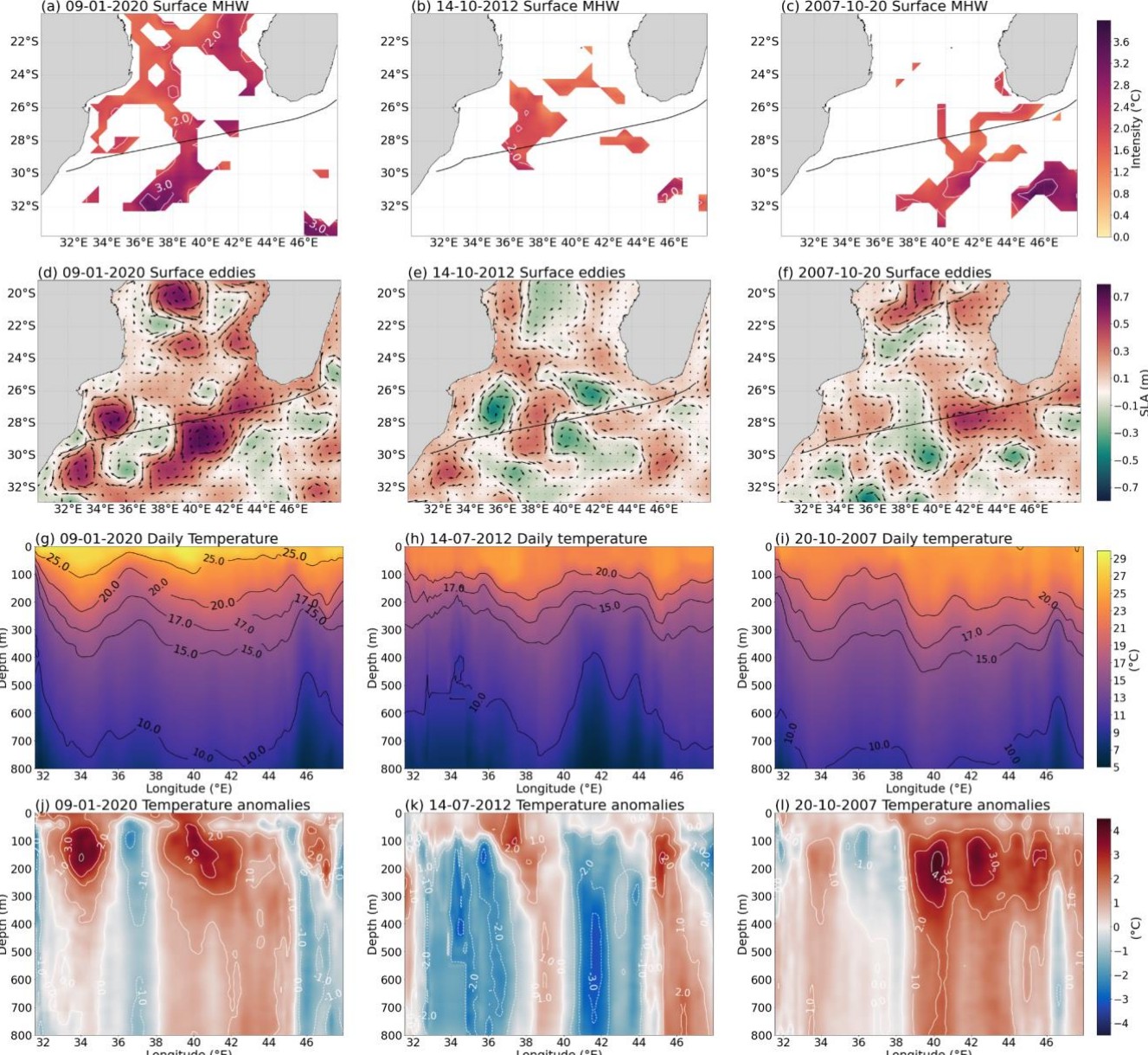

**Figure 6:** Surface MHW intensity using OISST V2 data (°C) that took place on **(a)** 9 January 2020, **(b)** 14 July 2012, **(c)** 20 October 2007. Sea level anomalies (m) and geostrophic current velocities (m/s) on the three MHW case study days **(d)** 9 January 2020, **(e)** 13 July 2012 and **(f)** 20 October 2007 using optimally interpolated AVISO altimetry satellite data. The black solid line indicates the position of the XBT transect (IX21). Subsurface temperature **(g – i)** and temperature anomalies **(j – l)** from the 9th – 13th January 2020 **(g and j),** the 14th – 17th July 2012 **(h and k)** and the 20ᵗʰ – 24th October 2007 **(i and l).** The subsurface profiles are from *in situ* XBT data from the IX21 transect line (see methods section).

Moreover, anomalously warm temperatures surrounding warm–core eddies are present with depth at the locations where MHWs persist over the XBT transect line (Fig. 6a–c and g–l). During the January 2020 MHW event, two distinct columns of anomalously warm temperatures (1–4 °C) extend from the surface to 800 m at 34° E and 38–45° E (Fig. 6j). At 36–38° E, the location of the surface–identified July 2012 MHW, anomalously warm water (1–2 °C) extends from the surface to 800 m (Fig. 6h). Between 39–46° E, the October 2007 MHW has temperature anomalies (1–4 °C) that extend below the surface–identified MHW to 800 m (Fig. 6l).

The subsurface temperature anomalies associated with surface–identified MHWs, intensify below the surface (Fig. 6a,b,c,j,k,l and Table 1). During the January 2020 MHW case study, the subsurface temperature anomalies, relative to the surface MHW intensity (2.5 °C) at 34° E and between 38–45° E, are most extreme from 50–200 m, reaching up to 4 °C (Fig. 6a and j and Table 1). Similarly, subsurface temperature anomalies during the October 2007 MHW event reach maximum temperatures (4°C) above the seasonal mean below 100 m at 40° E, 42° E and 45° E (Fig. 6l and Table 1). Although much weaker, at 38° E, the subsurface temperature anomaly that extends below the surface–identified July 2012 MHW has a maximum anomaly of 2 °C that extends from 0–150 m (Fig. 6k and Table 1). Given that the mixed layer depth, on average, in this region does not exceed 100 m (de Boyer Montégut et al., 2004), the most intense MHW temperatures are experienced below the mixed layer, and not at the surface.

**Table 1:** Surface and subsurface MHW characteristics for all three case studies: location of surface MHW over XBT transect (Location), temperature anomalies (maximum surface intensity ('Max. Surface'), maximum temperature anomaly ('Max. subsurface') and the depth where it occurs ('Depth of Max')), SLA at the location of surface–identified MHWs (SLA) where positive indicates the presence of a warm core eddy.

| MHW case studies | Location | Temperature anomalies | | | SLA (m) |
|---|---|---|---|---|---|
| | | Max. Surface (°C) | Max. Subsurface (°C) | Depth of Max. (m) | |
| 09–01–2020 | 34° E; 38–45° E | 2.5 | 4 | 100–200 | 0.5 |
| 14–07–2012 | 36–38° E | 2 | 2 | 100 | 0.2 |
| 20–10–2007 | 39–46° E | 1.5 | 4 | 100–200 | 0.4 |

However, weaker subsurface temperature anomalies are associated with warm–core eddies where there is no surface identified MHW signal (Fig. 6a–l). At 46° E, during both the January 2020 and July 2012 MHW case studies, a weak subsurface temperature anomaly (1–2°C) extends below the surface where no surface MHW was identified, but a warm–core mesoscale

eddy (SLA of 0.2–0.4 m) is found (Fig. 6 a,b, j and k). Similarly, between 32 – 36° E during the October 2007 MHW, a weak
column of anomalously warm water (1–2 °C) extends from 0–800 m and is associated with an eddy (SLA of 0.3 m) rather than
a surface identified MHW (Fig. 6c, f and l). Nonetheless, the most extreme subsurface temperature anomalies strongly reflect
the vertical extent of the surface MHW, rather than a warm core eddy signature, as the anomalies associated with surface
identified MHWs and mesoscale eddies are larger than those associated with mesoscale eddies alone (Fig. 6a–c and g–l).
Overall, the three MHW case studies have different spatial distributions and intensities, but there are several commonalities
between all three events (Fig. 6). All three case studies indicate that MHWs are associated with warm–core anticyclonic eddies
and subsurface temperature anomalies that extend down to at least 800 m and intensify below the surface typically within the
thermocline.
**4. Discussion**
Our study reveals distinct spatial patterns in the surface and subsurface extent of MHW characteristics within the SWIO and
emphasizes the critical role of mesoscale eddies in shaping both surface and subsurface anomalies. The subsurface
intensification of surface–identified MHW signals, which is modulated by mesoscale processes, highlights the importance of
understanding the subsurface dynamics when assessing MHW impacts and underscores the importance of an integrated
approach to studying MHW variability in this region.
On average, the range of the mean annual surface MHW metrics (frequency, intensity, duration and cumulative intensity) are
considered typical for highly dynamic WBCs systems and are classified as intense and abrupt events (Oliver et al., 2018;
Holbrook et al., 2019; Marin et al., 2022). However, across the SWIO, MHW metrics exhibit distinct spatial distributions
which are linked to the underlying patterns of SST variability and mesoscale eddy activity. These regional differences of MHW
characteristics are likely driven by the local processes that drive SST variability and the variable circulation itself (Oliver et
al., 2018).
In particular, areas of low SST variance are characterized by more frequent, less intense and shorter events, whereas areas of
high SST variance have less frequent MHW events but they are more intense and longer lasting.  The regions of low SST
variance fall within the main passageway of the eddies through the MC and from the SEMC leakage into the AC source region.
Here, less intense but frequent MHWs are expected and found in our study, as these are typical characteristics of MHWs in
WBCs where the high abundance of rapidly propagating eddies is known to dictate the growth and decay of MHWs (Frölicher
et al., 2018; Oliver et al., 2018; Oliver, 2019; Spillman et al., 2021; Fragkopoulou et al., 2023). Since the IX21 XBT transect
runs through the region of  high eddy activity, our results capture the subsurface anomalies associated with MHW events that
are influenced by mesoscale eddy activity and occur frequently, but have reduced annual duration, intensity and cumulative
intensity, compared to the rest of the region.
Moreover, mesoscale eddies play a critical role in shaping the depth of subsurface temperature anomalies associated with
surface MHWs. Distinct columns of deep, anomalously warm temperatures were located below surface–identified MHWs.

The majority of the subsurface anomalies associated with MHWs extend to depths down to 800m, with maximum temperature anomalies occurring beneath the surface. Surface temperature anomalies are correlated with their associated maximum subsurface temperature anomalies as well as the depth of the subsurface anomalies, meaning that more intense MHWs experience deeper and warmer maximum subsurface anomalies. Specifically, anti–cyclonic warm–core eddies are associated with much deeper maximum subsurface anomalies. This is observed during all three case studies, where the MHWs persist within anti–cyclonic eddies that vary in magnitude. The more intense 2007 and 2020 MHWs experience much stronger and deeper maximum subsurface temperature anomalies, associated with stronger anti–cyclonic eddies, compared to the 2012 MHW subsurface temperature anomaly, associated with a weaker anti–cyclonic eddy. This further demonstrates that, as is seen in other WBCs, anomalously warm anticyclonic eddies may act as a mechanism for MHW intensification below the surface (Schaeffer and Roughan, 2017; Elzahaby et al., 2021; Mawren et al., 2022 a; Azarian et al., 2024) (Schaeffer and Roughan, 2017; Elzahaby and Schaeffer, 2019; Elzahaby et al., 2021; Perez et al., 2021; Amaya et al., 2023; Fragkopoulou et al., 2023; Azarian et al., 2024). Furthermore intensification of warm–core eddies, under the current WBC warming trends, may subsequently amplify MHW events and intensify their subsurface signals in the future (Wu et al., 2012; Schaeffer and Roughan, 2017; Benthuysen et al., 2020). This growing influence of mesoscale eddies on subsurface MHWs, under ongoing global warming, may be driven by the faster response of the mixed layer compared to the slower response of the deep ocean, which affects the vertical distribution of heat and emphasizes the value of considering vertical structures in future ocean warming studies (Zhang et al., 2023; Azarian et al., 2024; He et al., 2024).

Since regions with high eddy abundance are likely to experience a large proportion of deep, subsurface–intensified MHWs, the exposure of vital coastal ecosystems to these events may be significantly underestimated when MHWs are studied using satellite data alone (Elzahaby & Schaeffer, 2019; Fragkopoulou et al., 2023). This underestimation is particularly concerning in the SWIO, where MHWs have previously been linked to severe coral bleaching events, such as the extreme bleaching of Le Grand Récif de Toliara, one of the largest and most biodiverse barrier reef systems (Mawren et al., 2022a). In addition, the co–occurrence of MHWs and tropical cyclones in the SWIO exacerbates their impacts, as cyclones can intensify MHW events, leading to more severe thermal stress. These combined phenomena have had devastating consequences for coastal marine ecosystems, particularly off southeastern Africa, where frequent and intense MHWs have triggered severe coral bleaching (Mawren et al., 2022b). Damage to habitat–forming coral species initiates cascading effects on marine ecosystems, threatening biodiversity and fisheries that are essential to the livelihoods of coastal communities in Madagascar and surrounding regions (Obura, 2012; Pereira et al., 2014; Obura et al., 2021; Mawren et al., 2022b). This highlights the importance of examining subsurface MHW signals, as satellite–based analyses alone may overlook the full extent of the thermal stress and ecological consequences for mesopelagic fishes, including changes in prey availability for deep–diving predators (Iglasias et al., 2024).

It should be noted, however, that although mesoscale eddies do influence the subsurface characteristics of anomalies associated with MHWs, past heat budget studies of global MHWs have shown that abrupt and intense MHWs are likely driven by a combination of processes, such as local advection, eddy heat flux, air–sea heat flux and large–scale climate modes, which control heat variations over various spatio–temporal scales (Hayashida et al., 2020; Marin et al., 2022). The complexity of the

SWIO therefore suggests that it is likely a combination of different oceanic and atmospheric processes that drive MHWs
and warrant further investigation. Ultimately, this will enhance our understanding of subsurface biological impacts and inform
management strategies aimed at preventing irreversible damage to ocean ecosystems..

**5. Conclusions**

To date, our study is the first to describe the subsurface extent of MHWs in the Greater Agulhas Current System, the WBC of
the SWIO. Furthermore, the distinct spatial patterns in the surface and subsurface extent of MHW characteristics emphasize
the critical influence of mesoscale eddies in shaping these thermal anomalies. The strong relationship between surface MHW
signals and subsurface temperature anomalies highlights the importance of considering subsurface dynamics when assessing
MHW impacts. By analyzing case studies of persistent MHW events, we demonstrate how mesoscale processes modulate both
the intensity and vertical structure of these anomalies, underscoring the need for an integrated approach to studying MHW
variability in this region This finding demonstrates that investigation of MHWs using only surface satellite data may
significantly underestimate the severity and impacts of MHWs. The presence of anticyclonic warm–core eddies influences
both the surface distribution and subsurface signals of MHWs, in particular their depth extent.
Our results highlight the need for consistent and long-term subsurface data to better understand the progression, frequency,
and duration of MHWs below the surface. Investigating heat budgets and the drivers of MHWs will provide deeper insights
into the mechanisms behind these events, enhancing the accuracy of forecasting models and improving management strategies
to mitigate the impacts on marine ecosystems.
Future research should prioritize specific locations within the SWIO where MHWs are most extreme or frequent, focusing on
areas with high marine biodiversity, such as coral reef systems. These targeted studies will be crucial for understanding local
MHW characteristics and their effects on fisheries and coastal economies, ultimately contributing to more effective
conservation and resource management efforts.

**Code availability**

The code used to detect MHWs is available at https://github.com/ecjoliver/marineHeatWaves.

**Data availability**

All data used in this study are open access. The daily NOAA OISST V2 data is available
at https://coastwatch.pfeg.noaa.gov/erddap/ (Huang et al., 2020). The HR-XBT data is made available by the Scripps
Institution of Oceanography HR-XBT program (IX21 - http://www.hrx.ucsd.edu/ix15.html) The AVISO product is available
from CMEMS (https://doi.org/10.48670/moi-00148, E.U. CMEMS).

## Author contributions

CBW performed the analysis and led the writing of the manuscript at University of Cape Town, South Africa. JCS, JS, TM, DM and NM supervised, proposed and guided the project, and contributed to the writing and analysis.

## Competing interests

The authors declare that they have no conflict of interest.

## Disclaimer

## Acknowledgements

The research leading to these results has received funding from the National Research Foundation (NRF) through Grant PMDS230630125138, the South African Environmental Observation Network (SAEON) and the University of Cape Town. JS was supported by the NOAA Global Ocean Monitoring and Observing Program through Award NA20OAR4320278. NM was supported by Australian Research Council Future Fellowship FT220100475.

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
