# Peer review of "Subsurface manifestation of Marine Heatwayes in the South West"

_EGUsphere, 2024_

## Author Response (AR1)

**Response to reviewer 1:**
**Manuscript number: egusphere-2024-2210**
**Manuscript title: Subsurface manifestation of Marine Heatwaves in the South West Indian Ocean**

Dear reviewer,

We sincerely thank Reviewer 1 for your valuable feedback and constructive suggestions, which have significantly improved the quality and clarity of our manuscript. We have carefully addressed all comments, including the addition of quantitative analyses to strengthen the link between EKE, SST variability, and MHW metrics, as well as clarifications on MHW metrics, time periods, and data selection. Additionally, we have incorporated minor text edits and updates to figure captions and titles to enhance readability and precision. We believe these revisions address all concerns raised and contribute to a more robust and comprehensive study on the characterization and impacts of MHWs in the SWIO. We appreciate your thorough review and are confident that the revised manuscript now provides a clearer and stronger contribution to the field.

To make the responses easier to follow, we have colour coded our responses as well as reviewer comments using the key below:

- **In bold black: Reviewer 1**

- In plain blue: our responses

**General comments:**

**The paper focuses on the characterization of Marine Heat Waves (MHW) at the surface and subsurface in the Southwest Indian Ocean region and the role that mesoscale eddies might play in their generation. The results are relevant in terms of addressing knowledge gaps on the vertical extent of temperature extremes and for regional impacts on biodiversity and fisheries. The paper is well-written, and the methods are sound overall. However, the link between EKE, SST and MHW metrics present in Fig 3 is quite qualitative. Adding a spatial correlation analysis, for example, would strengthen the suggested links between eddies and MHW activity.**

Thank you for your insightful feedback and for recognizing the value of our work in addressing knowledge gaps regarding the vertical extent of MHWs in the Southwest Indian Ocean and their potential links to mesoscale eddies.

We agree that the relationship between Eddy Kinetic Energy (EKE), Sea Surface Temperature (SST), and MHW metrics in Figure 3 could be further quantified. To further statistically quantify these relationships a scatterplot was generated with MHW intensity plotted against SST standard deviation, and a regression line was fitted to the data. The results reveal a statistically significant positive correlation ($r = 0.75$, $p < 0.001$), indicating that regions with higher SST variability are associated with increased MHW intensity. To further strengthen the link between these variables, a colour bar representing EKE was overlaid on the scatterplot. The results demonstrate that regions with higher EKE values tend to cluster in areas with lower MHW intensity and SST variability, whereas lower EKE values tend to cluster ins

areas with higher MHW intensity and variability. This supports the hypothesis that mesoscale eddies play a significant role in modulating both SST variability and MHW activity. We have incorporated this new analysis and updated the manuscript to include the scatterplot and its interpretation (see revised Fig. 3 and Section 3.1). This additional analysis provides quantitative evidence to support the relationship between EKE, SST variability, and MHW metrics.

**It is also unclear why MHW total days (annual total of extreme days) and maximum intensity metrics were omitted from the analysis.**

In Figure 2, we show the annual mean MHW frequency (average no. of events per year), annual mean MHW duration (average no. of days per year), annual mean MHW intensity (°C) and annual mean MHW cumulative intensity (°C days) and these figure panels are discussed in the results section (lines 147-171). For consistency, we chose to show only the annual mean metrics. We computed both the suggested metrics of MHW total number of extreme days and the maximum intensity and found they did not provide any additional insight to the MHW characteristics and so for simplicity chose to omit them.

**Finally, the analysis time periods of SST, subsurface temperature and SLA, currents data are not consistent without a clear justification (see details in specific comments below). The authors need to address these points and the specific comments below.**

Thank you for bringing this to our attention. We have responded to the specific comments below and edited the manuscript accordingly. We hope this provides sufficient clarification and justification of the time periods selected.

**Specific comments:**

**Line 16: Explain what is meant by moderate MHW. Are they moderate in intensity or have moderate occurrences, ...?**

Moderate MHWs are considered abrupt, yet intense, which is typical of MHWs that occur in western boundary currents (Marin et al. 2022). Since this is not a generally accepted metric to describe MHWs, we have changed it to 'abrupt and intense'.

Edited text: Line 17 'abrupt and intense MHWs'

**Line 53: add 'the' before focus**

Edited text: Line 65 'the' added before focus

**Line 53-54: Can also add that the SWIO is one of the main 6 hotspots of global marine biodiversity (Ramirez et al. 2017; https://www.science.org/doi/10.1126/sciadv.1601198), which makes assessing MHW in this region even more crucial and urgent.**

Thank you for this recommendation. This is a useful and impactful addition to the paper.

Edited text: Line 65 – 69

'Here, the focus is on the South West Indian Ocean (SWIO), which is part of the greater western Indian Ocean global warming hotspot (Roxy et al., 2014), and is host to a highly

unique, complex and variable WBC current system that supports one of the six primary global marine biodiversity hotspots (Ramirez et al., 2017). This makes assessing MHWs in this region even more crucial and urgent, yet, to date, MHW characterization in the SWIO is sparse and remains limited to the surface (Mawren et al., 2022 a).'

**Line 96 section 2.1: Why the study time period was restricted to 1993-2022 when the satellite SST record extend from 1982 to 2023 (as full years)?**

While satellite SST data is available from 1982 – 2023, the XBT IX21-HR transect only has data available from 1993 – 2022. Thus we used the overlapping period 1993-2022 for both the SST and HR-XBT data sets. A 30 year climatological period (1993-2022) is a generally acceptable time period to reliably compute robust MHW metrics (Hobday et al., 2016; Smith et al., 2025)

Edited text: **Line 127 - 129**

'For consistency with previous studies, a fixed climatological baseline, 1993 – 2022, and a 31 day smoothing window was used to identify surface MHWs (Smith et al., 2025).'

**Line 102-103: What's the exact climatological period used in from what year to what year. Please specify.**

The climatological period used to detect MHWs was from 01/01/1993 – 31/12/2022. This is previously stipulated at line 97. To make it clearer, we have added the years from which the fixed climatological baseline was calculated at lines 101 and 109.

Edited text: **Line 119 - 120**

'High resolution gridded (0.25°) NOAA optimally interpolated sea surface temperature (OI SST) V2 data was used to explore SST conditions and identify surface MHWs in the SWIO region from 01/01/1993 – 31/12/2022'

**Line 127 - 129**

'For consistency with previous studies, a fixed climatological baseline, 1993 – 2022, and a 31 day smoothing window was used to identify surface MHWs (Smith et al., 2025).'

**Line 104-106: Why Total Days (annual total of extreme days) and maximum intensity metrics were omitted? Is there a specific reason?**

As noted above, we primarily considered only the mean annual frequency, intensity, cumulative intensity, and duration as MHW metrics in our analysis. We examined the total days and total maximum intensity metrics for the entire period and they show similar MHW characteristics and spatial distributions as the selected metrics, and so we do not think that the addition of these figures add any vital or unique insights for the purpose of this paper.

**Lines 128-129: Why just 20 years period only from 1993-2012. Please explain the choice of not using an SLA time period that extends to the end of the subsurface or surface temperature data (2022) when the aim is to examine the influence of eddies on MHWs?**

Thank you for bringing this to our attention. The sea level anomaly and geostrophic currents data was downloaded for the full period from 1993 – 2022, however the SLA data downloaded from AVISO altimeter data has already been computed with respect to a twenty-year (1993 – 2012) mean. To make this distinction and provide clarity, the manuscript has been edited accordingly.

Edited text: **Line 159 – 162:**

'To investigate the influence of mesoscale eddies on the properties of surface-identified MHWs, high resolution (0.25°), optimally integrated, gridded daily sea level anomalies (SLA) and geostrophic currents, over a thirty-year period (01/01/1993 – 31/12/2022), were extracted from altimeter satellite data distributed by AVISO (Archiving, Validation, and Interpretation of Satellite Oceanographic data). Altimeter satellite gridded SLA are computed with respect to a twenty-year [1993, 2012] mean.'

**Figure 2e title and caption: Do you mean "mean intensity"? If so, please update accordingly.**

Thank you for bringing this to our attention. Figure 2e represents the annual mean intensity. To provide clarity, we have updated the figure caption.

Edited text: **Line 180**

'Surface mean annual MHW (c) frequency, (d) duration (days), (e) mean intensity (ºC) and (f) cumulative intensity (ºCdays)'

**Figure 2 panels titles: To avoid confusion between SST and MHW metrics, please update the title as follows: b) SST standard deviation, c) MHW Frequency, d) MHW Duration, e) MHW Mean Intensity, f) MHW Cumulative Intensity**

Thank you for bringing this to our attention. We have updated Figure 2 panel titles to distinguish between SST and MHW metrics as per your recommendation. The updated titles are as follows: b) SST standard deviation c) MHW Frequency d) MHW Duration, e) MHW Mean Intensity, f) MHW Cumulative Intensity.

**Line 144-145: How were the seasonal MHW derived?**

MHW events over the 30 year climatological period were grouped according to the season in which they occurred to construct Seasonal MHWs - seasons were considered as: Summer: December, January and February; Autumn: March, April and May; Winter: June, July and August and Spring: September, October and November. The mean annual cumulative intensity MHW metric was considered for the seasonal metrics as it best describes the severity of the events. We have updated the manuscript to include a short description of how seasonal MHWs were derived in the methods section.

Edited text: **Line 132 -135**

'Seasonal MHW patterns were also investigated using the metric cumulative intensity, which provides a good description of the severity of MHW events (Mawren et al., 2021). Detected MHW events over the entire climatological mean were grouped by season (Summer -

December, January and February; Autumn - March, April and May; Winter - June, July and August and Spring - September, October and November).'

**Line 163: true but excluding MHW frequency.**

Agreed, however we discuss frequency separately in lines 227 - 229.

**Lines 191-195: This fits better in Discussion.**

Thank you for this suggestion. We agree and have moved these lines to the discussion.

Line 191 - 195 moved to discussion, now line 460 - 463.

**Line 204-205: The dates in Figure 4 captions do not always match those in panels titles. Correct accordingly.**

Thank you for pointing this out. We have carefully reviewed the dates in this figure's captions and the corresponding panel titles. The necessary corrections have been made to ensure consistency between the dates in both the captions and the panel titles. The corrected dates are as follows: **(a, d, g, j)** 9 January 2020, **(b, e, h, k)** 14 July 2012, **(c, f, i, l)** 20 October 2007. Please note that, based on the request from the second reviewer to include additional statistical analyses, two additional figures have been added to the results under a new section 3.2. The case studies mentioned here, that were previously found in Figure 4 have been moved to section 3.3 and are now found in Figure 6.

**Response to reviewer 2:**
**Manuscript number: egusphere-2024-2210**

**Manuscript title: Subsurface manifestation of Marine Heatwaves in the South West Indian Ocean**

Dear Reviewer,

We would like to express our sincerest gratitude to the reviewer for the constructive feedback and suggestions. We appreciate the recognition of our study's value, particularly regarding the regional focus and the clarity of our figures. We also acknowledge the suggestion to strengthen our statistical analysis, and agree with the reviewer that further statistical analysis is critical for the strength of the paper (as also highlighted by the other reviewer). In response to this valuable suggestion, we have now included section 3.2 in the results section for the statistical analysis of subsurface warm anomalies associated with surface MHWs based on the full ensemble of XBT data.

We believe these enhancements significantly improve the manuscript's robustness and hope that the reviewer finds our revisions satisfactory. By incorporating the requested statistical analyses, clarifying our figures, and expanding the discussion to include recent literature, we have addressed the major and minor concerns raised. We are confident that these revisions have enhanced the scientific rigor, clarity, and overall quality of our work. We sincerely hope the reviewer finds the updated manuscript satisfactory.

Below, we provide point-by-point responses to the major and minor concerns raised, outlining the changes made to the manuscript accordingly. To make the responses easier to follow, we have colour coded our responses as well as reviewer comments using the key below:

- **In bold black: Reviewer 2**

- In plain blue: our responses

**Major Concern: Need for Statistical Analysis of Subsurface Warm Anomalies associated with surface-identified MHWs**

**The paper presents a study of subsurface characterization of MHWs, which in the past have received more attention in terms of surface properties. The study is performed in the South West Indian Ocean, whose choice is well motivated both in terms of available datasets (a dense and recurrent array of XBTs), regional dynamics, and potential impacts of subsurface warm anomalies. The paper has two main claims: that MHWs have strong subsurface warm anomalies below the mixed layer and that these are mainly associated to the presence of eddies. In general, I found the paper well written, with pertinent bibliographic references, clear figures, and solid structure. Together with some very minor issues listed at the end, there is however in my view a critical issue that prevents the paper to be acceptable in the present form.**

**A statistical, robust identification of subsurface warm anomalies associated to surface MHWs is the key result that the reader waits for, and the pillar over which the entire paper could stand or fall. Nevertheless, this part is developed only in terms of a visual inspection of four case studies (Fig. 4). These case studies are well described and useful,**

**because they show concretely the type of events encountered and the datasets. However, a consolidation of this part by a statistical analysis is necessary. This in fact is somehow announced, as the authors write "For each day where subsurface data was available and a MHW signal was present over the XBT transect, the surface MHW intensity was compared to daily subsurface temperature and anomaly profiles." (L199-200). For some reasons, the result of this comparison beyond the case studies is not shown (the Result Section ends with the case study analysis). The paper has largely the space for an in-depth extra analysis, having at the moment only 4 figures, with 2 of them serving for the context.**

**In conclusion, the paper should provide a statistically robust analysis based on the ensemble of the XBT dataset presented, if the authors want to support quantitatively the two main claims namely, (i) the presence of subsurface warm anomalies associated to surface MHWs, and (ii) the association of these anomalies to the presence of mesoscale eddies. Without this additional analyses (i.e., at least two extra figures) in my view the paper is not strong enough for publication.**

We fully agree with the reviewer's suggestion that a statistical analysis is required to strengthen our claims regarding the presence of subsurface warm anomalies and their association with mesoscale eddies. In response, we have conducted a comprehensive statistical assessment using the full dataset of 65 MHW occurrences identified along the IX21 XBT transect. The method used to conduct the statistical analyses has also been included in the methods section of the revised manuscript. This can be found at 'Section 2.3 Investigation of subsurface anomaly signals associated with surface MHWs' and includes two additional figures which aim to statistically support our two main claims, namely (i) the presence of subsurface warm anomalies associated with surface MHWs, and (ii) the association of these anomalies to the presence of mesoscale eddies.

The following modifications have been made and added to the manuscript in the results section 3.2:

1.   The association of surface MHWs with subsurface warm anomalies:

Figure 4 represents the statistical analyses we conducted to quantitatively support our first main claim that surface MHWs are associated with subsurface warm anomalies. Figure 5a shows the scatterplot of surface anomaly temperatures compared to the maximum warm subsurface anomaly temperature, and a regression line was fitted to the data. From this plot, we statistically compare surface and subsurface anomalies, demonstrating 'a significant relationship between the two (r = 0.70, p-value < 0.0001) (Figure 4a)'. This supports the claim that stronger surface anomalies generally correspond to more intense subsurface anomalies.

To further strengthen our claim that MHWs intensify below the surface, we compared the distribution of surface anomaly temperatures to the maximum subsurface temperature anomalies (below 20m). This is shown in Figure 4b, by means of boxplots comparing surface and subsurface warm anomalies and a two-sample t-test was conducted to determine whether there is a significant difference between surface and subsurface warm anomalies. These statistics confirm that 'subsurface anomalies are significantly warmer than surface anomalies, with a mean difference of 1.04°C (p-value < 0.001) (Figure 4b)'.

We hope that these additional results directly address the reviewer's request for a robust identification of subsurface warm anomalies associated with surface MHWs.

2. Influence of Mesoscale Eddies on Subsurface Anomalies:

To statistically support our other main claim that subsurface warm anomalies are associated with the presence of mesoscale eddies, we evaluate the relationship between maximum subsurface temperature anomalies and their depth in relation to sea level anomalies (SLA). This statistical analysis is present in Fig. 5. Fig. 5a is a scatterplot, with a regression line plotted to the data showing the relationship between maximum subsurface temperature anomalies (°C) and their corresponding depths (m), color-coded by Sea Level Anomalies (SLA, m). Positive SLA values (red) indicate anti-cyclonic eddies, while negative SLA values (blue) represent cyclonic eddies. 'A positive statistically significant relationship between maximum subsurface anomaly temperature is found, with a correlation of 0.5 (p-value < 0.0001), showing that the warmest subsurface temperatures are typically associated with greater depths (Fig. 5a). This highlights a positive association between the magnitude of the subsurface temperature anomaly and the depth at which it occurs.' The colour coded SLA further demonstrates that positive SLA anomalies (indicative of warm anti-cyclonic eddies) are associated with most of the data points, especially the warmest and deepest subsurface temperature anomalies, whereas subsurface anomalies associated with negative SLA are less common (indicative of cooler cyclonic eddies), and are not as warm or deep. This points to mesoscale eddies being the underlying physical mechanism that drives deeper and more intense subsurface maximum anomalies.

To further statistically quantify the association between mesoscale eddies and subsurface warm anomalies, the depth of maximum subsurface anomalies associated with cyclonic and anti-cyclonic eddies was compared. This statistical analysis is present in Fig 5b, which consists of boxplots comparing the depths of maximum subsurface anomalies between anti-cyclonic and cyclonic eddies. This analysis reveals that 'anti-cyclonic eddies (positive SLA) are associated with significantly deeper subsurface anomalies compared to cyclonic eddies (p-value < 0.0001), with mean maximum anomaly depths of 100 m and 51 m, respectively (Figure 5b).' This finding quantitatively supports our second main claim: that mesoscale eddies modulate the depth and intensity of subsurface warm anomalies.

By incorporating these new statistical analyses and visualizations (Figures 4 and 5), we have strengthened our argument with robust, ensemble-based evidence rather than relying solely on case studies. The case studies now follow after the statistical analyses, in section 3.3 and are presented in Fig 7. They provide visual, and specific examples of the typical surface MHW intensity and distribution, their associated subsurface warm anomalies and the influence of mesoscale eddies in this region. Furthermore, the results from the case studies further corroborate the statistics we present in section 3.2.

**Minor Issues**

**"Fig. 2C: If I understand correctly, what is shown is the N of events/year. Please modify 'events' in 'events/year' to be consistent."**
The y-axis label in Figure 2C has been updated to "events/year" to improve clarity and maintain consistency.

**"I found just one more recent paper that also studied the subsurface signal of MHWs in the South Indian Ocean: Azarian et al. (2024). The authors may (or**

**may not) want to discuss their findings in its respect."**

Thank you for this reference. We have now cited and briefly discussed Azarian et al. (2024) in the discussion section, comparing our findings with theirs. Their study similarly identifies subsurface intensification of MHWs in the Indian Ocean sector of the Southern Ocean, aligning with our results but differing in regional focus and dataset used. The reference list has also been updated accordingly.

Summary of Revisions

We have made substantial improvements in response to the reviewer's concerns, including:
1. Conducting a full statistical analysis of subsurface anomalies, confirming their prevalence and strength.

2. Establishing a strong correlation ($r = 0.70$) between surface and subsurface temperature anomalies.

3. Demonstrating the significant difference between surface and subsurface anomalies using a t-test ($p < 0.001$).

4. Quantifying the influence of mesoscale eddies on subsurface anomaly depths, with anti-cyclonic eddies driving deeper anomalies ($p < 0.0001$).

5. Revising Figure 2C for clarity and

6. Including a recent relevant study (Azarian et al., 2024) in the discussion.

---

## Author Response (AR2)

**Responses to Reviewer 2: Minor Corrections**

**Manuscript number: egusphere-2024-2210**

**Manuscript title: Subsurface manifestation of Marine Heatwaves in the South West Indian Ocean**

**● In bold black: Reviewer 2**

● In plain blue: our responses

**I congratulate the authors for the new version of the manuscript, in particular for the subsurface statistical analysis. The manuscript has greatly improved and in my view there are only a few outstanding issues before recommending publication. I list them below by line number (track changes version) and indicate two of them as important. I recommended «minor revision » because addressing the issues indicated below should be relatively quick and does not require necessarily new analysis or extensive discussion. However, without correctly addressing these points (in particular the two important issues of L257) the paper in my view is not ready for publication.**

Dear Reviewer,

Thank you for your thoughtful feedback on the revised manuscript. We greatly appreciate your recognition of the improvements made, particularly with regard to the subsurface statistical analysis.
We have carefully addressed the outstanding minor issues and addressed each point carefully - including the two important concerns on line 257. Please note all changes list below by line are done so according to the line numbers from the track changes version of the revised manuscript.
We believe these revisions adequately address your concerns, and we look forward to your feedback. Thank you again for your valuable input.

**L18 « these eddy corridors » : this expression to me is not very clear, especially for an abstract. A corridor is a narrow and long passage. In the context of the abstract its meaning is not unique, as it can be intended as some space between several eddies, in the vertical across individual eddies, along a narrow region at the basin scale where eddy presence is higher, etc. I would suggest to provide in the text a compact definition or ading one or two keywords to specify at least at which scale this corridor has to be intended.**

In response to your suggestion, we have replaced the term "eddy corridors" with the phrase "regions of high eddy activity in the Mozambique Channel and southeast of Madagascar" to provide a clearer and more precise description. This revision specifies the geographical region of interest and avoids ambiguity regarding the scale or type of eddy feature. We hope this improves the clarity of the abstract.

Edited text: Was Line 18, Now Line 16
'Regions of high eddy activity in the Mozambique Channel and southeast of Madagascar.'

**L60 « WBCs current system » : « C » meaning already « current », maybe « WBC system » ?**

Thank you for noting this error. We have corrected it.

Edited text: Was Line 60, Now Line 55
'WBC system'

**L61 Ramirez et al. not in the bibliography**

Thank you for bringing this to our attention. The Ramirez et al. 2017 reference has now been added to the bibliography.

Edited text: line 567
Ramírez, F., Afán, I., Davis, L. S., and Chiaradia, A.: Climate impacts on global hot spots of marine biodiversity, Sci. Adv., 3, e1601198, https://doi.org/10.1126/sciadv.1601198, 2017.

**L108, 145 and 193 : High resolution (0.25°). Today, I would not call a product at 0.25° resolution « high resolution ». High resolution altimetry corresponds more to the resolution of the novel SWOT satellite (about 2km) and high resolution SST to infrared products (in the km range). Instead of « high resolution » I would suggest the term « mesoscale resolving ».**

We agree that the term "mesoscale resolving" is more appropriate for describing a 0.25° resolution in the context of modern altimetry and SST products. We have revised the text to replace "high resolution" with "mesoscale resolving" to better align with current terminology and to accurately convey the scale of the data.

Edited text: Was Line 108, 145 and 193,  Now Line 99, 135, 195, 221 and  223
Replaced 'high resolution' with 'mesoscale resolving'.

**L161-163 : subsurface warm temperature signal : I agree that the anomaly is likely to be warm in the presence of a surface marine heatwave, however, since this is a hypothesis to test, I would replace « subsurface warm temperature signal » with « subsurface anomaly temperature signal », so that no a priori assumption of a warm character of the anomaly is done in the Methods part.**

Thank you for this suggestion. We have replaced "subsurface warm temperature signal" with "subsurface temperature anomaly" in the Methods section to avoid assuming the nature of the anomaly. This revision ensures that no a priori assumption of a warm character of the anomaly is done in the Methods section.

Edited text: Was Line 161 - 163, Now line 151:
"Subsurface warm temperature signal" replaced with "subsurface temperature  anomaly"

**L254-256 : What are the two percentages for non-MHW profiles ?**

Non-MHW profiles were not included in our analysis, as our focus was specifically on understanding and characterising the depth extent of MHW events. Therefore, we do not provide corresponding percentages for non-MHW profiles. Investigating non-MHW profiles is beyond the scope of this study and would require additional statistical analysis which we do

not think would enhance the results of our study significantly enough to justify the additional analysis required.

**L257 (important issue): I am confused by the meaning of « subsurface » . Looking at figure 4a the large majority of profiles in MHWs have a maximum at 800m or deeper (they are dark blue), which looks to me surprisingly high. Can you confirm that the colorbar is correct? In this case, this is not consistent with your case studies analysis, in which there is no a single example of temperature anomaly maxima at 800m or higher. Something is wrong here, or possibly I am missing something.**

Thank you for bringing this to our attention. It seems there has been a misunderstanding of what Fig 4a represents and we apologise for this confusion. The colour bar for Fig 4a does not indicate the depth of the maximum subsurface temperature anomaly, but rather the depth extent of the positive temperature anomaly that extends from the surface MHW signal down the water column. This confusion has brought to our attention that we also did not describe how this was identified in the Methods Section.

To provide clarity, we have stipulated in the methods section 2.4 that for each temperature anomaly profile (associated with surface-identified MHWs), the depth extent of the positive anomaly in the water column was identified. We have specifically defined both the depth extent of the positive temperature anomaly and the depth of the maximum warm subsurface temperature anomaly, as both are investigated. In accordance with this edit, we have updated Fig 4a's colourbar heading as well.

Furthermore, we acknowledge that the wording of the paragraph starting L257 may have contributed to this confusion and have refined the paragraph to provide a clearer and more precise interpretation of Fig. 4a. We hope this resolves the confusion and appreciate the opportunity to clarify. Please note that some of the text presenting changes made for this comment also include changes made in response to the next comment regarding the consideration of subsurface anomalies only when they persist below the mixed layer depth. This is further explained in more depth in response to the next comment.

Edited text: Line 161 - 163  (Methods Section 2.4)
"... three metrics for subsurface MHWs were identified:  (1) the maximum depth extent (m) of the warm anomaly; (2) the maximum subsurface temperature anomaly (°C) and (3) the corresponding depth (m) of this maximum subsurface temperature anomaly."

Line 231 – 243 (Results Section 3.2)

"For each day and location where a MHW signal was detected over the XBT transect, all but two events exhibited warm temperature anomalies that extended from the surface down to the climatological MLD (47.15 m), with 68 % of them reaching down to 800 m. Furthermore, 80.25 % of the events experienced maximum temperature anomalies below the climatological MLD (47.15 m). Given that anomalies are considered subsurface if they extend below the MLD, these results suggest that the majority of surface-identified MHWs were associated with deep-reaching, subsurface-intensified warm anomalies. A significant relationship was found between the surface temperature anomalies and the maximum subsurface temperature anomalies ($r = 0.67$, p–value $< 0.0001$), as subsurface anomalies tend to increase with increasing surface anomalies (Fig. 4a). Furthermore, deeper MHW events, where the **maximum depth extent of the warm anomaly was greater than 500m**, tend to have warmer subsurface maximum temperature anomalies than shallower events. On average, events that extend deeper in the water column have surface temperature anomalies of 1.17°C and

maximum subsurface anomalies of 2.27°C. Whereas shallower events, where the **maximum depth extent of the warm anomaly was less than 500m**, have, on average, surface temperature anomalies of 1.18°C and maximum subsurface temperature anomalies of 1.91°C. This observed depth–dependent pattern suggests that the depth extent of the subsurface temperature anomalies may play a role in modulating subsurface thermal responses to surface anomalies."

Line 259 - 260 (Fig 4 Figure caption)
"Figure 4: a) Scatter plot of surface temperature anomalies and maximum subsurface temperature anomalies where surface MHWs were identified. Colours indicate the maximum depth extent of the warm anomaly (m)."

**L257 (important issue): The definition of « subsurface » should include only values starting below the mixed layer (even a rough estimation, like its climatological depth) in order to avoid to call « subsurface » a maximum that is just below the surface, and therefore may be higher than the surface value just because of noise. Please at least check that all the « subsurface » maxima are really subsurface, that is, their depth is at least below the mixed layer, and indicate this in the text.**

Thank you for your careful consideration of the definition of "subsurface." We fully agree that only maxima occurring below the mixed layer should be classified as "subsurface". To ensure consistency, we calculated the mean climatological mixed layer depth (47.15 m). The maximum subsurface temperature anomalies were then noted below this mixed layer depth. This adjustment guarantees that all identified subsurface anomalies are truly below the mixed layer and not influenced by near-surface noise. The manuscript has been updated accordingly, with a clear definition included in the text (Methods Section 2.4 and in Results Section 3.2) and the Fig. 4 and Fig. 5 have been updated to only consider subsurface anomalies below 47.15 m.

Edited text: Line 152 - 160 (Methods Section 2.4)

"Warm temperature anomalies extending below surface-identified MHWs, were only considered subsurface if they extended below the mean climatological mixed layer depth (MLD).
The climatological MLD was calculated using the temperature threshold method, similar to the approach by De Boyer Montégut et al. (2004) and Elzahaby and Schaeffer (2009). The MLD was determined as:
$$MLD = \min \{z \mid T(z) < T(10) - 0.2° C\} \tag{2}$$
where $T(z)$ represents temperature at depth z, and $T(10)$ is the reference temperature at 10 m depth. The threshold temperature (0.2° C) and reference depth (10m) were chosen based on the recommendations of De Boyer Montégut et al. (2004). The final climatological MLD was obtained by averaging MLD values across the study region, providing a robust estimate of 47.15 m."

Edited text: Line 231 – 235

"For each day and location where a MHW signal was detected over the XBT transect, all but two events exhibited warm temperature anomalies that extended from the surface down to the

climatological MLD (47.15 m), with 68% of them reaching down to 800 m. Furthermore, 80.25% of the events experienced maximum temperature anomalies below the climatological MLD (47.15 m). Given that anomalies are considered subsurface if they extend below the MLD, these results suggest that the majority of surface-identified MHWs were associated with deep-reaching, subsurface-intensified warm anomalies."

**L258-259 « Furthermore, deeper.. are more comparable (Fig. 4a). » I don't see this quantitatively in Fig 4a. I have the impression that there are more « green/yellow » dots below the red lines, but a quantification of your statement (for instance max anomaly average for the maxima deeper than 500 m vs the average of the ones shallower than 500m) should be provided.**

We appreciate your recommendation and agree that the addition of a quantitative comparison would further substantiate our statement. To address this, we have now included a quantification of the maximum subsurface temperature anomalies that extend below 500 m versus those temperature anomalies that are shallower than 500 m. This was quantified by determining the average surface and average maximum subsurface temperature anomaly for those anomalies that extend down to 500m (shallow events) and those temperature anomalies that extend between 500 – 800m depth. This analysis confirms that deeper subsurface anomalies are generally associated with warmer maximum subsurface anomalies, supporting our original statement. Please see our above response for the updated text and additional quantification.

Edited text: Line 238 – 243

"Furthermore, deeper MHW events, where the **maximum depth extent of the warm anomaly was greater than 500m**, tend to have warmer subsurface maximum temperature anomalies than shallower events. On average, events that extend deeper in the water column have surface temperature anomalies of 1.17°C and maximum subsurface anomalies of 2.27°C. Whereas shallower events, where the **maximum depth extent of the warm anomaly was less than 500m**, have, on average, surface temperature anomalies of 1.18°C and maximum subsurface temperature anomalies of 1.91°C. This observed depth–dependent pattern suggests that the depth extent of the subsurface temperature anomalies may play a role in modulating subsurface thermal responses to surface anomalies."

**Fig 5b : Again, I would consider « subsurface » only maxima that are deeper than the mixed layer.**

Noted. We have also updated Fig. 5 to ensure that only maxima occurring below the mixed layer depth (set at 47.15 m, as explained in our response to L257) are considered "subsurface". This revision ensures consistency across the analysis and avoids misclassification of near-surface anomalies. The figure and corresponding text in the manuscript have been edited to account for this edit.

**L389 : « The majority of the subsurface anomalies associated with MHWs extend to depths down to 800m, with maximum temperature anomalies occurring beneath the surface. » I don't think you showed that in general the majority of subsurface anomalies extend down to 800m. This has been shown only for three case studies. Unless I'm wrong,**

**please correct this statement. I also noticed that in between the warm anomalies there are important cold anomalies. This suggest an alternating pattern of heat waves / cold spells. The authors may or may not want to elaborate on this point in the Discussion.**

In results section 3.2 "Subsurface temperature anomaly properties associated with surface–identified MHWs" (line 232) we find that 68% of the warm subsurface anomalies have a maximum depth extent of 800m, also shown in Fig 4a, where the colour bar represents the 'maximum depth extent of the warm anomaly' for each temperature profile. We believe this is enough to justify our claim that, in general, the majority of the subsurface anomalies associated with MHWs extend to 800m.

Thank you for your observation of the alternating pattern of heatwaves/cold spells, which we agree is indeed an interesting characteristic of this region. However, the focus of this study is specifically on the warm anomalies associated with MHWs and, as such, cold spells were not investigated as part of this study. Since the main aims of this study were to characterise the subsurface extent of MHWs in the SWIO, we have chosen not to mention this observed alternating pattern. Cold spells in the Agulhas are the subject of our ongoing research.

**References : Azarian et al. 2024 : I mention this paper in my previous review, however the authors have added Azarian et al. 2023. Azarian et al. 2023 talks about surface heat waves, the 2024 paper I was referring to (https://doi.org/10.1016/j.jmarsys.2023.103962) about some subsurface signature of marine heatwaves. This is a minor issue, the authors may leave as is now, or update the discussion with the 2024 paper.**

Thank you for pointing this out. We cited the incorrect reference in the bibliography. We have now revised the manuscript to include the correct reference, Azarian et al. (2024), which aligns with the discussion on subsurface signatures of marine heatwaves.

Edited Text: Line 440 – 442

"Azarian, C., Bopp, L., Sallée, J.-B., Swart, S., Guinet, C., and d'Ovidio, F.: Marine heatwaves and global warming impacts on winter waters in the Southern Indian Ocean, Journal of Marine Systems, 243, 103962, https://doi.org/10.1016/j.jmarsys.2023.103962, 2024."